# Mechanical Ventilation during Extracorporeal Membrane Oxygenation in Acute Respiratory Distress Syndrome: A Narrative Review

**DOI:** 10.3390/jcm10214953

**Published:** 2021-10-26

**Authors:** Li-Chung Chiu, Kuo-Chin Kao

**Affiliations:** 1Department of Thoracic Medicine, Chang Gung Memorial Hospital, Chang Gung University College of Medicine, Taoyuan 33305, Taiwan; pomd54@cgmh.org.tw; 2Graduate Institute of Clinical Medical Sciences, College of Medicine, Chang Gung University, Taoyuan 33302, Taiwan; 3Department of Respiratory Therapy, Chang Gung University College of Medicine, Taoyuan 33302, Taiwan; 4Department of Respiratory Therapy, Chang Gung Memorial Hospital, Chang Gung University College of Medicine, Taoyuan 33305, Taiwan

**Keywords:** acute respiratory distress syndrome, extracorporeal membrane oxygenation, mechanical ventilation, ventilator-induced lung injury, multiple organ failure

## Abstract

Acute respiratory distress syndrome (ARDS) is a life-threatening condition involving acute hypoxemic respiratory failure. Mechanical ventilation remains the cornerstone of management for ARDS; however, potentially injurious mechanical forces introduce the risk of ventilator-induced lung injury, multiple organ failure, and death. Extracorporeal membrane oxygenation (ECMO) is a salvage therapy aimed at ensuring adequate gas exchange for patients suffering from severe ARDS with profound hypoxemia where conventional mechanical ventilation has failed. ECMO allows for lower tidal volumes and airway pressures, which can reduce the risk of further lung injury, and allow the lungs to rest. However, the collateral effect of ECMO should be considered. Recent studies have reported correlations between mechanical ventilator settings during ECMO and mortality. In many cases, mechanical ventilation settings should be tailored to the individual; however, researchers have yet to establish optimal ventilator settings or determine the degree to which ventilation load can be decreased. This paper presents an overview of previous studies and clinical trials pertaining to the management of mechanical ventilation during ECMO for patients with severe ARDS, with a focus on clinical findings, suggestions, protocols, guidelines, and expert opinions. We also identified a number of issues that have yet to be adequately addressed.

## 1. Introduction

Acute respiratory distress syndrome (ARDS) is characterized by acute respiratory failure with severe hypoxemia. Lung-protective mechanical ventilation strategies with lower tidal volumes and lower airway pressures remain the mainstay of ARDS management aimed at improving survival [1,2]. However, mechanical ventilation could potentially cause ventilator-induced lung injury (VILI) and contribute to non-pulmonary organ failure, increasing the risk of mortality [3,4,5,6].

Extracorporeal membrane oxygenation (ECMO) is a rescue therapy aimed at improving gas exchange for severe ARDS patients with life-threatening hypoxemia refractory to conventional mechanical ventilation [7,8,9,10]. Advances in extracorporeal support techniques may improve outcomes for patients with severe ARDS, including those with coronavirus 2019 [11]. Two recent randomized controlled trials, Conventional Ventilatory Support versus Extracorporeal Membrane Oxygenation for Severe Adult Respiratory Failure (CESAR) [12] and the ECMO to Rescue Lung Injury in Severe ARDS (EOLIA) [13], reported that ECMO has potential survival benefits for patients with severe ARDS. Unfortunately, neither study was able to detect meaningful between-group differences in survival, nor have various ethical and methodological issues been noted [14]. One meta-analysis based on data from individual patients (including the CESAR and EOLIA trials) determined that 90-day mortality was significantly lower in the ECMO group than in groups undergoing conventional management [14].

ECMO allows an ultra-protective ventilation to lower tidal volumes and airway pressures aimed at mitigating VILI [7,8,15,16]. Recent studies have reported that mechanical ventilator settings during ECMO may have an impact on mortality in patients with severe ARDS [17,18,19,20,21,22]. Nonetheless, researchers have yet to conduct large-scale, prospective, randomized controlled trials investigating optimal mechanical ventilator settings, the extent to which ventilator load can be reduced during ECMO, or the effect of ventilator settings on clinical outcomes among patients with severe ARDS.

This review article examined previous studies, clinical trials, and organization protocols pertaining to mechanical ventilator settings in patients with severe ARDS undergoing venovenous (VV) ECMO, focusing on clinical findings, recommendations, guidelines, and expert opinions.

## 2. Ventilator-Induced Lung Injury

### 2.1. Introduction

VILI results from interaction between mechanical ventilation and lung parenchyma, and is caused by excessive mechanical stress or strain to the lung parenchyma (Figure 1a). Both mechanical ventilation and applied lung pathophysiology/mechanics should be taken into account for the causes of VILI [3,6].

### 2.2. Ventilator-Related Causes of VILI

The adverse effects of mechanical ventilation in cases of ARDS can be categorized as follows: (1) a non-physiological increase in transpulmonary pressure; (2) a non-physiological increase or decrease in pleural pressure during positive or negative pressure ventilation. The primary adverse effects associated with excessive transpulmonary pressure and pleural pressure are VILI and hemodynamic alterations, respectively [6]. Mechanical ventilator settings attributable to VILI include volume, pressure, flow, and respiratory rate. These factors together determine the energy load transmitted to the lung parenchyma per unit of time, which is referred to as mechanical power (MP) [23]. The same MP derived from different extents of volume, pressure, flow, and respiratory rate could cause diverse effects on the respiratory system.

### 2.3. Pathophysiology of the Lung Parenchyma Associated with VILI

The individual pathophysiological conditions associated with the occurrence of VILI include functional lung size, the extent of lung inhomogeneity, and lung recruitability [3,6]. Thus, increasing lung homogeneity or recruitability may decrease the risk of VILI. Note that the effects of energy load (i.e., MP) depend largely on the condition of the lung parenchyma (i.e., the same MP applied to different conditions of the lung parenchyma could cause diverse effects), which is generally assessed using computed tomography scans [24]; however, the implementation of ECMO may preclude such assessments.

## 3. Multiple Organ Failure Due to VILI

The most common cause of death among ARDS patients is multiple organ failure [1,2]. Damage to the alveolar epithelium due to VILI prompts the release of numerous proinflammatory cytokines and chemokines, which can translocate into the circulatory system leading to distal organ dysfunction and death, in a process referred to as biotrauma (Figure 1a) [4,5].

ECMO is meant to limit mechanical forces and thereby may prevent VILI, biotrauma, and the risk of mortality. One randomized controlled trial reported that significant reductions in plateau pressure, tidal volume, and driving pressure by VV ECMO significantly reduce the risk of pulmonary biotrauma in patients with severe ARDS, compared with standard protective-lung ventilation (as indicated by cytokine concentrations) [25].

Recent studies reported that extrapulmonary organ failure during ECMO was significantly correlated with mortality among patients with severe ARDS [17,26,27,28]. ECMO can improve gas exchange to reduce the risk of tissue hypoxemia. It can also facilitate a reduction in ventilator load (i.e., MP) delivered to alveoli to alleviate VILI by reducing the concentration of pulmonary and systemic inflammatory mediators and thereby may decrease the risk of multiple organ failure (Figure 1b) [5,14,17,27].

## 4. Mechanical Ventilator Settings during ECMO

### 4.1. Introduction

ECMO allows for a reduction of tidal volumes and airway pressures (i.e., reducing the loads on the lungs) to mitigate the risk of further VILI [7,8]. Nonetheless, researchers have yet to establish optimal strategies pertaining to ventilation intensity during ECMO. Note that this is due largely to the need to tailor settings to the respiratory mechanics of the individual. Table 1 lists recent studies that investigated the effects of mechanical ventilator settings on the clinical outcomes of ARDS patients with ECMO. Table 2 summarizes the initial mechanical ventilation parameters for ARDS patients with ECMO in recent clinical trials, organization protocols, and expert opinions. Table 3 summarizes all aspects of ventilation settings during the initial phase of ECMO for patients with ARDS.

### 4.2. Modes of Mechanical Ventilation

No previous studies have compared the modes of ventilation during ECMO in terms of clinical outcomes. Many patients are deeply sedated and paralyzed during the initial phase of ECMO involving pressure- or volume-controlled ventilation. Pressure-controlled modes allow the daily monitoring of tidal volume increase as a function of lung compliance or clinical condition improving. As a result, pressure-controlled modes are widely preferred in the initial phase of ARDS, as indicated by their inclusion in Extracorporeal Life Support Organization (ELSO) guidelines [12,15,30].

One international multicenter prospective cohort study reported that 50% of ARDS patients received volume-targeted ventilation, whereas 40% received pressure-targeted ventilation prior to ECMO. Note, however, that the usage of pressure-targeted modes increased with the duration of ECMO, as follows: day 1 (69%), day 7 (71%), and day 14 (82%) [26]. The EOLIA trial focused on volume-assisted ventilation or airway pressure release ventilation [13]. Airway pressure release ventilation refers to pressure controlled, intermittent mandatory ventilation that uses two levels of airway pressure (high pressure and low pressure). It is typically applied using inverse inspiratory-expiratory ratios with unrestricted spontaneous breathing. This strategy has been shown to improve oxygenation (compared with conventional ventilation modes); however, it does not appear to provide a significant advantage in terms of clinical outcomes [31].

### 4.3. Mechanical Power

MP refers to the amount of energy per unit of time transmitted to the lung parenchyma during mechanical ventilation, and probably contributes to the development of VILI and clinical outcomes. MP is derived from the volume, pressure, flow, and respiratory rate. Thus, it is reasonable to assume that MP is superior to individual ventilator parameters in predicting the risk of VILI [23].

Researchers have yet to define a safe MP threshold for patients with critical illnesses with or without ARDS. One experimental model reported the occurrence of lung edema and lung damage when MP exceeded a threshold of 12 J/min [32]. High MP levels have also been independently associated with an elevated risk of in-hospital mortality in critically ill patients, and researchers have reported a consistent increase in in-hospital mortality when MP is increased beyond 17 J/min [33]. In a standardized screening study, MP values exceeding 22 J/min have been associated with increased 28-day hospital mortality and 3-year mortality in ARDS patients [34].

ECMO provides ultra-protective ventilation by reducing airway pressure and energy load (i.e., MP) transmitted to the lungs, which can potentially promote lung healing and mitigate further lung injury. There is currently no clearly defined threshold for MP during ECMO by which to predict outcomes for patients with severe ARDS. One recent study reported that the 90-day hospital mortality was significantly higher among patients with high mean MP (>14.4 J/min) during the first 3 days of ECMO, compared to patients with low mean MP (≤14.4 J/min) (70.7% versus 46.8%, *p* = 0.004) [17].

A given energy load (i.e., MP) could have a diversity of effects on the respiratory system depending on the pressure, volume, and respiratory rate as well as the individual pathophysiology of the lungs, such as the functional lung size, the extent of inhomogeneity, and recruitability [3,6]. Theoretically, the functional lung size (i.e., remaining aerated lung) of patients with severe ARDS requiring ECMO is smaller than that of patients with mild or moderate ARDS. At the same time, the inhomogeneity and lung recruitability are both greater, thereby increasing the risk of VILI in severe ARDS patients [17,24,35]. This means that MP should be adjusted for functional lung size (at the very least) to reflect the amount of energy load delivered to the lungs.

Specific power is defined as power per ventilated lung unit or power referenced to the dimensions of the ventilated lung. It is assumed that specific power enables more accurate predictions of VILI [36,37]. In a recent study, it was found that MP during the first 3 days of ECMO was the only ventilatory variable independently associated with 90-day hospital mortality, and MP referenced to compliance during ECMO was more predictive for mortality than was MP alone (adjusted HR 2.289, *p* = 0.010 versus adjusted HR 1.060, *p* = 0.005, respectively) [17].

### 4.4. Tidal Volume

Lung-protective ventilation strategies that lower tidal volume (4–8 mL/kg predicted body weight, PBW) to reduce stress and strain on the lungs have survival benefits for ARDS patients. Typically, the tidal volume is reduced to as low as 4 mL/kg PBW in cases where the plateau pressure exceeds 30 cmH_2_O [1,31,38].

For severe ARDS patients supported with ECMO, tidal volumes are often adjusted with the aim of achieving plateau pressure ≤ 24 cmH_2_O, and typically maintain below 4 mL/kg PBW as ultra-protective ventilation during ECMO to minimize the risk of VILI [7,13,27]. One prospective international multicenter study reported that tidal volumes were significantly reduced from 6.4 ± 2.0 mL/kg PBW at ECMO initiation to 3.7 ± 2.0 mL/kg PBW during the first 2 days of ECMO support (*p* < 0.001) [26]. It is unclear whether further reductions in tidal volume could further improve outcomes.

### 4.5. Positive End-Expiratory Pressure

Positive end-expiratory pressure (PEEP) is the pressure used to maintain the alveolar opening during end expiration. Higher PEEP levels are meant to increase mean airway pressure to improve oxygenation, reduce tidal lung stress and strain, maintain alveolar recruitment, and prevent alveolar collapse at end expiration to decrease the risk of lung inhomogeneity, VILI, and intrapulmonary shunt [35,39].

The potential harmful effects of PEEP include increased pleural pressure, elevated right atrial pressure, and a reduced pressure gradient for venous return, which can contribute to decreased cardiac output. PEEP has also been shown to increase pulmonary vascular resistance, which elevates right ventricular afterload and could further reduce cardiac output [38,39].

The optimal PEEP is generally estimated using gas exchange (PEEP/FiO_2_ tables, dead space), respiratory mechanics (compliance, driving pressure, pressure–volume curve, stress index, esophageal manometry), and/or imaging studies (electrical impedance tomography, ultrasonography, chest computed tomographic imaging) [31]. However, one recent randomized clinical trial reported that among moderate to severe ARDS patients, PEEP titration guided by esophageal pressure did not provide a significant benefit over an empirical high PEEP-FiO_2_ strategy in terms of mortality or days free from mechanical ventilation [40]. A recent study revealed that PEEP titration guided with electrical impedance tomography, compared with pressure–volume curve, might be associated with improved driving pressure and survival rate in patients with moderate to severe ARDS [41].

ECMO allows for lower tidal volume ventilation (typically below 4 mL/kg PBW), which may contribute to atelectasis and severe ventilation/perfusion mismatch unless PEEP is appropriately increased to keep part of the lung open [15]. Unfortunately, higher PEEP during ECMO tends to inhibit venous return and negatively affects hemodynamics in cases involving VV ECMO [30]. When applying ECMO to ARDS patients, PEEP should be set according to the alveolar recruitability, pleural pressure, and hemodynamics of the individual [27]; however, this is not necessarily feasible in clinical practice.

Researchers have yet to determine the optimal PEEP target during ECMO; however, one study suggested a value of ≥10 cmH_2_O [27]. ELSO guidelines recommended the value of PEEP can be as high as tolerated and avoid inhibition of venous return [30]. One retrospective study reported that higher PEEP during the first 3 days of ECMO was independently associated with lower mortality (OR 0.75, *p* = 0.0006) [20].

### 4.6. Plateau Pressure

Plateau pressure is defined as the pressure obtained at end-inspiration after a 0.5 s inspiratory pause when patients are sedated and paralyzed. Mechanical ventilation using lower inspiratory pressures (e.g., plateau pressure < 30 cmH_2_O) has been strongly recommended for patients with ARDS [1,38,42].

ECMO is considered a rescue therapy for severe ARDS patients with profound hypoxemia or uncompensated hypercapnia; i.e., those who are unable to tolerate the excessively high inspiratory airway pressure of conventional mechanical ventilation [7]. Ultra-protective ventilation is meant to limit plateau pressure ≤ 24 cmH_2_O [13,27] or peak inspiratory pressure at 20–25 cmH_2_O [12].

One systematic review summarized ventilation practices for ARDS patients with ECMO. The results reported that after ECMO support, plateau pressure was decreased by a median of 4.3 cmH_2_O (3.5–5.8) and mortality was lower among patients who had lower intensity of applied ventilation following ECMO initiation [16]. Another international prospective study reported that plateau pressure is generally reduced from 32 ± 7 cmH_2_O at the time of ECMO initiation to 24 ± 7 cmH_2_O during the first 2 days of ECMO (*p* < 0.001) [26]. One cohort study of patients with influenza A (H1N1)-induced ARDS receiving ECMO reported that higher plateau pressure on the first day of ECMO was independently associated with higher ICU mortality. They concluded that outcomes could be improved by implementing ultra-protective ventilation with target tidal volumes aimed at minimizing plateau pressure [18].

### 4.7. Driving Pressure

Driving pressure refers to the difference between the plateau pressure and PEEP, which is inversely proportional to respiratory system compliance. Respiratory system compliance is correlated with the amount of aerated lung tissue available for tidal ventilation (i.e., functional lung size or the dimension of ARDS baby lung) in patients with ARDS [36,43]. It is reasonable to adjust the tidal volume or PEEP to minimize driving pressure.

One post hoc observational study of 3562 patients with ARDS in nine randomized controlled trials reported that for patients with ARDS, driving pressure is the ventilation variable with the greatest predictive value for mortality [43]. However, the causal relationship between driving pressure and outcomes has not been confirmed, and higher driving pressure may be just another marker for ARDS severity.

Researchers have yet to establish a clearly defined safe upper limit for driving pressure during ECMO, and values below 15 cmH_2_O are defined as ultra-protective ventilation [26]. One international multicenter study reported a reduction in driving pressure from 20 ± 7 cmH_2_O at the time of ECMO initiation to 14 ± 4 cmH_2_O within the first 2 days of ECMO (*p* < 0.001) [26]. Other studies have reported that driving pressure during the first 3 days of ECMO was the only ventilator variable independently associated with hospital mortality [21,22]. One recent randomized controlled trial reported a linear relationship between changes in driving pressure and plasma concentrations of various inflammatory mediators during VV ECMO in patients with severe ARDS. They also reported that reducing driving pressure to zero during VV ECMO may be beneficial and provide more lung-protective ventilation [44]. However, further research is required to assess the benefits and risks of this approach.

### 4.8. Transpulmonary Pressure

Transpulmonary pressure refers to the difference between pressure inside the alveoli and pleural pressure, largely involved with distending the lung parenchyma. Esophageal manometry is the method used clinically to measure pleural pressure in assessing the pathophysiology and mechanics of the respiratory system, which consider the effect of chest wall [42,45]. At this point, however, its use was limited in clinical practice due to insufficient knowledge and technical difficulties.

Unlike plateau pressure and driving pressure, transpulmonary pressure reflects directly the individual physiology of the patient, and may therefore have a better impact on clinical outcomes and mortality in ARDS patients [10,46,47], particularly when dealing with obese patients [48].

One case series of patients with influenza A (H1N1)-associated ARDS supported with ECMO adjusted PEEP according to the upper physiologic limit of transpulmonary pressure (25 cmH_2_O) rather than the upper limit of plateau pressure (30 cmH_2_O). They reported that this approach improved oxygenation and offset the need for ECMO implementation [49].

One recent randomized controlled trial investigated the effect of transpulmonary pressure-guided mechanical ventilation on VILI in patients with severe ARDS treated using VV ECMO. Transpulmonary pressure guidance was more effective than a lung rest strategy in weaning patients from ECMO. It also resulted in significantly higher PEEP, lower driving pressure, lower tidal volumes, lower MP, and lower concentration of proinflammatory cytokines (interleukin-1β, 6, and 8) over time [50]. It appears that individually titrated mechanical ventilation based on transpulmonary pressure is safe and beneficial for patients with severe ARDS receiving ECMO.

### 4.9. Respiratory Rate

Lung-protective ventilation aimed at reducing tidal volumes has been shown to provide survival benefits for ARDS patients; however, it often brings with it the risk of respiratory acidosis, even in cases where the respiratory rate is increased. Note also that a higher respiratory rate has been shown to cause lung damage, even under lower tidal volumes. Respiratory rate is usually adjusted to maintain the partial pressure of carbon dioxide in arterial blood (PaCO_2_) within an acceptable range. A few studies have examined the effect of respiratory rate on VILI and clinical outcomes. The LUNG SAFE study reported a link between increased respiratory rate and increased hospital mortality in patients with ARDS [51].

The effect of respiratory rate on MP is less pronounced than that of tidal volume, driving pressure, and inspiratory flow [23]; however, one experimental study reported that an increase in mechanical power resulting from an increase in respiratory rate could induce lung edema and damage [32]. Recent studies have reported that after ECMO, respiratory rate decreased more precipitously than did the other determinants of MP [17,26,52]. One study recommended a target respiratory rate of 10 breaths per minute or less (as in the CESAR trial [12]) during ECMO for ARDS [27]. ELSO guidelines suggest a respiratory rate of only 5 breaths per minute [30].

### 4.10. Fraction of Inspired Oxygenation

VV ECMO is widely used to promote gas exchange in order to improve arterial oxygenation in patients with severe ARDS. Systemic arterial saturation of roughly 80% is typical during VV ECMO support. High initial blood flow in the extracorporeal circuit can be lowered to maintain arterial saturation of 80–85% [30]. The fraction of inspired oxygen (FiO_2_; measured at the ventilator) should be maintained at a low level to reduce the risk of oxygen toxicity and reabsorption atelectasis [15]. Some studies have suggested setting FiO_2_ at 0.3–0.5 [13,27]. Nonetheless, the objective should be to ensure systemic oxygen delivery rather than maintain a particular saturation level [27,30].

## 5. Uncertainties and Future Research

### 5.1. Introduction

ECMO gives the lungs a chance to rest; however, researchers have yet to establish evidence-based guidelines pertaining to optimal mechanical ventilation strategies during ECMO. Table 4 lists ongoing clinical trials evaluating the effectiveness of mechanical ventilation strategies during ECMO. In the following, we examine issues that remain unresolved at this point.

### 5.2. Spontaneous Breathing during ECMO

Spontaneous breathing can have protective or deleterious effects, depending on the severity of lung injury, the strength of spontaneous activity, respiratory patterns, patient-ventilator dyssynchrony, and the phase and duration of ARDS [27]. In cases of severe ARDS, vigorous spontaneous effort can promote lung injury via increased transpulmonary pressures and transmural pulmonary vascular pressure (i.e., patient self-inflicted lung injury; P-SILI) [27,54].

Few ARDS patients are able to tolerate ECMO strategies based on spontaneous breathing, due largely to the fact that patients requiring ECMO support are most severe with high respiratory drive. During the initial phase of ECMO support, researchers advise neuromuscular blockade to eliminate patient–ventilator dyssynchrony and thereby mitigate the risk of P-SILI [55,56]. Sedatives and paralytic agents can have detrimental effects; therefore, practitioners must seek a balance between minimizing sedation and reducing the risk of VILI.

However, titrating the amount of extracorporeal carbon dioxide removal could influence and control respiratory drive during ECMO and may allow spontaneous breathing in select patients with ARDS [27]. In patients recovering from severe ARDS undergoing pressure support ventilation and neurally adjusted ventilatory assist, lower tidal volume, lower peak airway pressure, and lower transpulmonary pressure were found when extracorporeal carbon dioxide extraction was increased and the PaCO_2_ levels decreased [57]. This indicated that spontaneous breathing during ECMO seemed to be feasible for select patients in the recovery phase of ARDS and may decrease the risk of VILI. The effects of extracorporeal carbon dioxide removal on spontaneous breathing still needs large randomized controlled trials in the future to be investigated.

### 5.3. Apneic and Near-Apneic Ventilation

Researchers have yet to determine whether near-apneic or apneic ventilation during ECMO (i.e., very low or zero respiratory rate) could be used to decrease the intensity of mechanical ventilation and thereby minimize the risk of VILI and biotrauma.

In one experimental model, patients with severe ARDS supported with ECMO who underwent nonprotective ventilation for 24 h (PEEP 5 cmH_2_O, tidal volume 10 mL/kg, respiratory rate 16–20 breaths per minute) exhibited severe histologic lung injury with early fibroproliferative response. Note that near-apneic ventilation (PEEP 10 cmH_2_O, driving pressure 10 cmH_2_O, respiratory rate 5 breaths per minute) was more effective than conventional protective ventilation (PEEP 10 cmH_2_O, tidal volume 6 mL/kg, respiratory rate 20 breaths per minute) in reducing this response [58].

Another randomized clinical trial concluded that in patients with severe ARDS, the use of continuous positive airway pressure without cyclic stress/strain (i.e., zero respiratory rate, no tidal ventilation) under ECMO may be the optimal ventilation strategy to minimize the risk of VILI (reflected by plasma cytokine concentrations) [44].

Near-apneic or apneic ventilation strategies require that patients be sedated and paralyzed without spontaneous breathing to prevent patient–ventilator dyssynchrony and P-SILI; however, this can lead to atelectasis. Maintaining apneic or near-apneic ventilation may require total ECMO dependent with a high blood flow rate to ensure adequate oxygenation. Note that researchers have yet to establish how long this approach (extreme lung protection) can feasibly be administered in clinical practice [59].

A very low respiratory rate can be used to reduce the ventilation load transmitted to lung parenchyma and thereby prevent lung injury; however, it is important to weigh the benefits against the risks. One of the studies mentioned above employed an animal model of ARDS and the other study enrolled a small sample of ARDS patients (*n* = 10). Large randomized controlled trials will be required to confirm the efficacy of this approach in terms of clinical outcomes.

### 5.4. Extracorporeal Carbon Dioxide Removal

Patients with ARDS receiving low tidal volume are prone to hypercapnia and corresponding respiratory acidosis. Extracorporeal carbon dioxide removal (ECCO_2_R) is used to reduce respiratory acidosis by clearing carbon dioxide; however, the low blood flow associated with this procedure (approximately 200–1500 mL/min) is insufficient to improve oxygenation. Nonetheless, ECCO_2_R makes it possible to reduce tidal volumes, airway pressure, and the respiratory rate [8,60,61,62,63].

ECCO_2_R has been shown to safely lower the respiratory rate and attenuate the expression of inflammatory mediators without altering respiratory mechanics, gas exchange, or hemodynamics in an experimental ARDS model [64]. One prospective multicenter study reported on the use of ECCO_2_R to permit ultra-protective ventilation in patients with moderate ARDS [65]. Another study reported that the lung-protective benefits of ECCO_2_R were more pronounced in cases with higher alveolar dead space fraction, lower respiratory system compliance, and high-performance ECCO_2_R devices [66].

Conversely, a recent multicenter, randomized clinical trial reported that among patients with acute hypoxemic respiratory failure, the use of ECCO_2_R to reduce tidal volume did not have a significant advantage over conventional low tidal volume ventilation in terms of 90-d mortality [53]. Note that the study was stopped early due to futility, and may have lacked the power required to detect clinically important differences.

Researchers have yet to determine whether ECCO_2_R could be applied in less severe cases of ARDS without life-threatening hypoxemia. Randomized controlled trials will be required to determine whether the overall benefits outweigh the drawbacks [63,67,68]. It is also important to remember that ECCO_2_R devices differ in terms of CO_2_ removal efficiency and potential adverse effects.

### 5.5. Weaning from ECMO and Mechanical Ventilation

It is possible that optimal ventilator settings during ECMO could promote lung healing and accelerate weaning from ECMO and mechanical ventilation. Weaning from ECMO can be initiated when clinicians observe improvements in lung infiltration, arterial oxygenation, and respiratory system compliance. In some cases, pressure-support mode ventilation may be preferred [7,62].

Nonetheless, researchers have yet to determine whether patients should be weaned from ECMO or the mechanical ventilator first. Overall, the decision depends on the condition of the patients; however, most intensivists prioritize ECMO weaning over mechanical ventilator weaning. In cases facing a higher risk of complications due to ECMO involving bleeding and/or hemolysis, decannulation may take precedence over extubation. Extubation may be given priority in cases of patient–ventilator dyssynchrony requiring substantial sedation and neuromuscular blockade, ventilator related barotrauma (like pneumothorax), or ventilator-associated pneumonia [27].

## 6. Collateral Effect of ECMO

Although there have been advances in supportive care and innovations in extracorporeal support techniques, there are many potential collateral effects or complications associated with ECMO and these can be lethal. Close daily monitoring to minimize the risk of ECMO-related complications is necessary, and requires intensive education and training [8]. Therefore, only hub centers have shown reduce mortality [69]. The adverse events of ECMO include mechanical (i.e., ECMO circuit or device) and medical complications.

The common complications directly related to the ECMO circuit are oxygenator failure, blood clots in oxygenator and other circuit, and other cannula or mechanical-related problems (e.g., vessel perforation or dissection, limb ischemia, and incorrect location of cannula). Bleeding is the most common complication during ECMO because of systemic anticoagulation and thrombocytopenia, and is one of the leading causes of mortality. Other complications not directly related to the ECMO circuit include thromboembolism, hemolysis, disseminated intravascular coagulation, culture-confirmed infection at any site, neurologic injury (e.g., rapid decrease in PaCO_2_), and renal failure [7,10].

## 7. Conclusions

ECMO provides ultra-protective ventilation in patients with severe ARDS, thereby allowing the lungs to rest, and may reduce the risk of progressive VILI and subsequent multiple organ failure. The collateral effects of ECMO may be devastating and should be investigated. Mechanical ventilation settings should be tailored to the individual; however, researchers have yet to establish optimal ventilator settings or determine the degree to which ventilation load can be decreased. Large-scale, prospective, randomized controlled trials are still required to answer these questions.

## Figures and Tables

**Figure 1 jcm-10-04953-f001:**
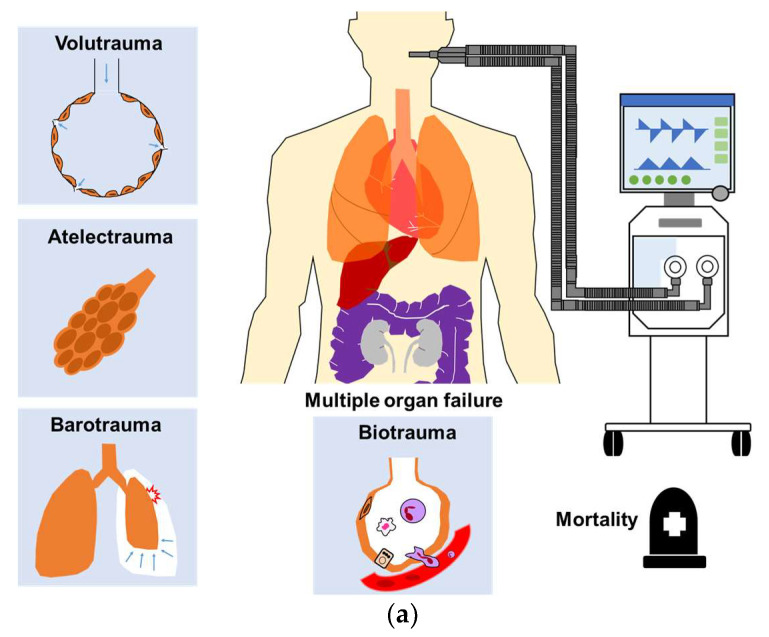
(**a**) Mechanical ventilation can cause VILI including volutrauma, atelectrauma, barotrauma, and biotrauma, which has been shown to contribute to multiple organ failure and mortality in patients with ARDS; (**b**) ECMO mitigates ventilation load to allow the lungs to rest, and may reduce the risk of VILI and multiple organ failure. (VILI, ventilator-induced lung injury; ARDS, acute respiratory distress syndrome; ECMO, extracorporeal membrane oxygenation; MP, mechanical power; V_T_, tidal volume; Pplat, plateau pressure; ∆P, driving pressure; RR, respiratory rate; FiO_2_, fraction of inspired oxygen).

**Table 1 jcm-10-04953-t001:** Overview of the recent studies investigating the impact of mechanical ventilator settings during ECMO on outcomes in patients with ARDS.

Studies	Study Design	Main Results
Pham et al., 2013 [18] (*n* = 123)	Retrospective observational study	ICU mortality: 35.8% Higher plateau pressure on the first day under ECMO was independently associated with ICU mortality (OR 1.33, *p* < 0.01)
Schmidt et al., 2015 [20] (*n* = 168)	Retrospective study	ICU mortality: 29% Higher PEEP during the first 3 days of ECMO was independently associated with lower mortality (OR 0.75, *p* = 0.0006)
Marhong et al., 2015 [16] (*n* = 2042)	Systematic review	Median (IQR) overall mortality: 41% (31–51%) Mortality was lower in patients receiving lower intensity of applied ventilation during ECMO. Combined tidal volume ≤ 4 mL/kg PBW and plateau pressure ≤ 26 cmH_2_O during ECMO had lowest mortality
Modrykamien et al., 2016 [19] (*n* = 64)	Retrospective observational study	Hospital mortality: 46.9% Increased plateau pressure was independently associated with decreased odds of hospital survival (OR 0.79, *p* = 0.007)
Neto et al., 2016 [21] (*n* = 545)	Individual patient data meta-analysis of observational studies	In-hospital mortality: 35.2% Driving pressure was the only ventilatory parameter during ECMO that was independently associated with in-hospital mortality (adjusted HR 1.06, *p* < 0.001)
Kim et al., 2019 [29] (*n* = 56)	Retrospective study	Hospital mortality: 48.1% Lung compliance during ECMO was significantly associated with 6-month mortality (HR 0.943, *p* = 0.009)
Schmidt et al., 2019 [26] (*n* = 350)	International prospective cohort study	Six-month mortality: 39% MV settings during the first 2 days of ECMO did not impact the prognosis
Chiu et al., 2021 [17] (*n* = 152)	Retrospective study	Hospital mortality: 53.3% MP during the first 3 days of ECMO was the only ventilatory variable independently associated with 90-day hospital mortality, and MP referenced to compliance had the greatest predictive value for mortality compared to MP alone (adjusted HR 2.289, *p* = 0.010)

ECMO: extracorporeal membrane oxygenation; ARDS: acute respiratory distress syndrome; ICU: intensive care unit; OR: odds ratio; PEEP: positive end-expiratory pressure; IQR: interquartile range; PBW: predicted body weight; HR: hazard ratio; MV: mechanical ventilation; MP: mechanical power.

**Table 2 jcm-10-04953-t002:** Initial mechanical ventilation settings during ECMO for patients with ARDS in clinical trials, organization, or a consensus of expert opinions.

Trials/Organization/Consensus	Mechanical Ventilator Settings
CESAR trial, 2009 [12]	Pressure-controlled ventilation Peak inspiratory pressure 20–25 cmH_2_O PEEP 10–15 cmH_2_O RR 10 breaths per minute FiO_2_ 0.3
ELSO guideline, 2017 [30]	First 24 h: moderate to heavy sedation Pressure-controlled ventilation 25 cmH_2_O, PEEP 15 cmH_2_O (PEEP can be as high as tolerated and avoid inhibition of venous return), plateau pressure < 25 cmH_2_O, inspiratory/expiratory ratio 2:1, RR 5 breaths per minute, FiO_2_ 0.5 After 24–48 h: moderate to minimal sedation Pressure-controlled ventilation 20 cmH_2_O, PEEP 10 cmH_2_O, Inspiratory/expiratory ratio 2:1, RR 5 breaths per minute plus spontaneous breaths, FiO_2_ 0.2–0.4 After 48 h: minimal to no sedation Pressure-controlled ventilation as above or CPAP 20 cmH_2_O plus spontaneous breathing Tracheostomy or extubation within 3–5 days
EOLIA trial, 2018 [13]	Volume-assist control mode: Plateau pressure ≤ 24 cmH_2_O Tidal volume lowered to obtain plateau pressure ≤ 24 cmH_2_O PEEP ≥ 10 cmH_2_O RR 10–30 breaths per minute FiO_2_ 0.3–0.5 Airway pressure release ventilation: High pressure ≤ 24 cmH_2_O PEEP ≥ 10 cmH_2_O RR 10–30 breaths per minute FiO_2_: 0.3–0.5
ECMONet expert opinions’ consensus conference, 2018 [27] *	Plateau pressure ≤ 24 cmH_2_O and may be lower if feasible Tidal volume: typically ≤4 mL/kg PBW, often much lower and adjusted for the goal of plateau pressure PEEP ≥ 10 cmH_2_O Driving pressure ≤ 14 cmH_2_O RR ≤ 10 breaths per minute FiO_2_: 0.3–0.5

ECMO: extracorporeal membrane oxygenation; ARDS: acute respiratory distress syndrome; CESAR: Conventional Ventilatory Support versus Extracorporeal Membrane Oxygenation for Severe Adult Respiratory Failure; PEEP: positive end-expiratory pressure; RR: respiratory rate; FiO_2_: fraction of inspired oxygen; ELSO: Extracorporeal Life Support Organization; CPAP: continuous positive airway pressure; EOLIA: ECMO to Rescue Lung Injury in Severe ARDS; PBW: predicted body weight. * Fourth Annual International ECMO Network Scientific Meeting in Rome, Italy, in 2018 (www.internationalecmonetwork.org/conferences, accessed on 22 July 2021).

**Table 3 jcm-10-04953-t003:** Summary of recommended ventilation settings during the initial phase of ECMO for patients with ARDS.

Mechanical Ventilator Settings	Target
Plateau pressure	≤24 cmH_2_O and may be lower if feasible
PEEP	≥10 cmH_2_O
Driving pressure	≤14 cmH_2_O
Tidal volume	Typically ≤4 mL/kg PBW and adjusted for the goal of plateau pressure (≤24 cmH_2_O)
Respiratory rate	≤10 breaths per minute
FiO_2_	0.3–0.5

ECMO: extracorporeal membrane oxygenation; ARDS: acute respiratory distress syndrome; PEEP: positive end-expiratory pressure; PBW: predicted body weight; FiO_2_: fraction of inspired oxygen.

**Table 4 jcm-10-04953-t004:** Overview of ongoing clinical trials pertaining to mechanical ventilation during ECMO for patients with ARDS.

Trial Names (Identifier, Status)	Inclusion Criteria	Interventional Group	Control Group	Primary Outcomes
New Lung Ventilation Strategies Guided by Transpulmonary Pressure in VV-ECMO for Severe ARDS (NCT02439151, published [50])	ARDS with reversible cause (PaO_2_/FiO_2_ < 80)	Transpulmonary pressure guide new lung ventilation strategy in ECMO for severe ARDS patients	Conventional ventilation strategy (ELSO guide ventilation strategy) in ECMO for severe ARDS patients	Proportion weaned from VV-ECMO
pRotective vEntilation With Veno-venouS Lung assisT in Respiratory Failure (REST) (NCT02654327, published [53])	Invasive MV within 48 h of acute potentially reversible hypoxemic respiratory failure (PaO_2_/FiO_2_ ≤ 150 mmHg) receiving PEEP ≥ 5 cmH_2_O	VV-ECCO_2_R: Plateau pressure ≤ 25 cmH_2_O Target tidal volume ≤ 3 mL/kg PBW	Standard care: Conventional lung protective mechanical ventilation	All causes of mortality at day 90
Low frequency, ultra-low tidal volume ventilation in patients with ARDS and ECMO (NCT03764319, recruiting)	Moderate to severe ARDS ECMO < 24 h in situ	Ultra-protective ventilator settings: Plateau pressure 23–25 cmH_2_O, tidal volume < 4 mL/kg PBW, PEEP 14–16 cmH_2_O, RR 4–5 bpm	Standard ventilator settings: Plateau pressure ≤ 35 cmH_2_O PEEP 8–12 cmH_2_O, RR 12–15 bpm	Ventilator-free days at day 28
Biomarkers, Genomics, Physiology in Critically Ill and ECMO Patients (IGNITE) (NCT04669444, enrolling by invitation)	Patient with ARDS supported with ECMO or a potential ECMO candidate A single interventional group	Low Driving Pressure Protocol: Initial driving pressure of 10–15 cmH_2_O and then decreased as tolerated for 2 h to evaluate the effects on pulmonary, cardiac, and inflammatory biomarkers	None	Change in plasma interleukin-6 level from baseline to low driving pressure ventilation
Ultra-Low Tidal Volume Mechanical Ventilation in ARDS Through ECMO (ULTIMATE) (NCT04832789, not yet recruiting)	Age ≥ 18 years Endotracheal mechanical ventilation ≤ 5 days Early moderate-severe ARDS	Ultra-protective ventilation with VV ECMO	Best conventional ventilation	Proportion of patients adhering to the study protocol and crossing over to VV ECMO Number of patients recruited for the study

ECMO: extracorporeal membrane oxygenation; ARDS: acute respiratory distress syndrome; VV: venovenous; PaO_2_: partial pressure of oxygen in arterial blood; FiO_2_: fraction of inspired oxygen; ELSO: Extracorporeal Life Support Organization; MV: mechanical ventilation; PEEP: positive end-expiratory pressure; ECCO_2_R: extracorporeal carbon dioxide removal; PBW: predicted body weight; RR: respiratory rate.

## Data Availability

All data will be available from the corresponding author on reasonable request.

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
