# Peer review of "Mechanical Ventilation during Extracorporeal Membrane Oxygenation in Acute Respiratory Distress Syndrome: A Narrative Review"

_jcm, 2021, doi:10.3390/jcm10214953_

Round 1
Reviewer 1 Report
I have read the review from Dr. Chiu and Dr. Kao with interest. This is a nicely organized review paper that summarized recent activities in ARDS patients using ECMO. I have a few suggestions.
- Title: this is not a systematic but rather narrative review, which should be clearly identifiable in the title.
- Section 2.3. The expression “lung-related causes of VILI” sounds a bit weird although I can understand what it means. Please consider to rephrase it.
- 1: The text in Volutrauma cannot be read. Either make it larger or delete it.
- Table 1: “multicenter” doesn't indicate the type of the study
- Table 1: “Systemic review ” should be “systematic review”
- Lines 215-222: as a controversial topic, you should also cite other studies that PEEP titration does lower the mortality (e.g. Hsu HJ et al. Physiol Meas. 42:014002).
- Section 5.2: You have clearly stated the issues with preserving spontaneous breathing and it seems not feasible during ECMO, so why do you think this would be a future research topic?
Author Response
I have read the review from Dr. Chiu and Dr. Kao with interest. This is a nicely organized review paper that summarized recent activities in ARDS patients using ECMO. I have a few suggestions.
Point 1: 
 Title: this is not a systematic but rather narrative review, which should be clearly identifiable in the title.
Response 1: We thank the reviewer to point out this problem and we agreed that this article is a narrative review. We revised the title in the revised manuscript as follows:
Mechanical Ventilation during Extracorporeal Membrane Oxygenation in Acute Respiratory Distress Syndrome: A Narrative Review
Point 2: Section 2.3. The expression “lung-related causes of VILI” sounds a bit weird although I can understand what it means. Please consider to rephrase it.
Response 2: We thank the reviewer’s suggestion to rephrase this expression. We revised the expression as “Pathophysiology of the Lung Parenchyma Associated with VILI” in page 2, the section 2.3. in the revised manuscript and marked with red words.
Point 3: 1: The text in Volutrauma cannot be read. Either make it larger or delete it.
Response 3: We thank the reviewer to point out this problem and we apologized for the unclear text in Volutrauma in Figure 1. We deleted the text in Figure 1 in the revised manuscript.
Point 4: Table 1: “multicenter” doesn't indicate the type of the study
Response 4: We thank the reviewer’s suggestion and we deleted the word “multicenter” in Table 1 in the revised manuscript.
Point 5: Table 1: “Systemic review” should be “systematic review”
Response 5: We thank the reviewer’s suggestion and we revised it as “systematic review” in Table 1 in the revised manuscript.
Point 6: Lines 215-222: as a controversial topic, you should also cite other studies that PEEP titration does lower the mortality (e.g. Hsu HJ et al. Physiol Meas. 42:014002).
Response 6: This is an excellent point of view. We appreciate with the reviewer’s comment to cite the study that PEEP titration guided with electrical impedance tomography, compared with pressure-volume loop, does lower the mortality in moderate to severe ARDS [1].
We added the description in the third paragraph of the section of “4.5. Positive End-expiratory Pressure” in the revised manuscript as follows (marked with red text):
A recent study revealed that PEEP titration guided with electrical impedance tomography, compared with pressure–volume curve, might be associated with improved driving pressure and survival rate in patients with moderate to severe ARDS [40].
We also added the reference 40.
Reference:
- Hsu, H.J.; Chang, H.T.; Zhao, Z.; Wang, P.H.; Zhang, J.H.; Chen, Y.S.; Frerichs, I.; Möller, K.; Fu, F.; Hsu, H.S.; et al,. Positive end-expiratory pressure titration with electrical impedance tomography and pressure-volume curve: a randomized trial in moderate to severe ARDS. Physiol Meas. 2021, 42, 014002.
Point 7: Section 5.2: You have clearly stated the issues with preserving spontaneous breathing and it seems not feasible during ECMO, so why do you think this would be a future research topic?
Response 7: We thank the reviewer’s comment to point out this issue. We agreed that spontaneous breathing during initial phase of ECMO for patients with severe ARDS may not be feasible, and we apologized for the lack of clarity. A previous study had revealed that it is difficult to apply a spontaneous breathing strategy during ECMO for patients in the early phases of severe ARDS [1].
However, the amount of extracorporeal carbon dioxide removal could influence and control respiratory drive in ARDS patients undergoing ECMO and may allow spontaneous breathing in select patients with ARDS [2].
A prospective, randomized study reported that select patients recovering from severe ARDS undergoing pressure support ventilation and neurally adjusted ventilatory assist (n = 8), lower tidal volume, lower peak airway pressure and lower transpulmonary pressure were found when extracorporeal carbon dioxide extraction was increased and the partial pressure of carbon dioxide in arterial blood (PaCO2) levels decreased [3]. It indicated that spontaneous breathing during ECMO seemed to be feasible for select patients in the recovery phase of ARDS and may decrease the risk of ventilator-induced lung injury (VILI).Therefore, titrating the amount of extracorporeal carbon dioxide removal to achieve acceptable respiratory drive and effort may decrease the risk of VILI in spontaneously breathing patients recovering from severe ARDS.
In conclusion, spontaneous breathing in ARDS patients receiving ECMO may be harmful when applied too early and remained to be investigated. However, a spontaneous breathing strategy during ECMO may be feasible and possibly limit the risk of VILI for select patients in the recovery phase of ARDS, and this strategy needs large randomized controlled trials to determine in the future.
We added the above description in the third paragraph of the section “5.2. Spontaneous Breathing during ECMO” in the revised manuscript as follows (marked with red text):
However, titrating the amount of extracorporeal carbon dioxide removal could influence and control respiratory drive during ECMO and may allow spontaneous breathing in select patients with ARDS [27]. In patients recovering from severe ARDS undergoing pressure support ventilation and neurally adjusted ventilatory assist, lower tidal volume, lower peak airway pressure and lower transpulmonary pressure were found when extracorporeal carbon dioxide extraction was increased and the PaCO2 levels decreased [55]. It indicated that spontaneous breathing during ECMO seemed to be feasible for select patients in the recovery phase of ARDS and may decrease the risk of VILI. The effects of extracorporeal carbon dioxide removal on spontaneous breathing still needs large randomized controlled trials in the future to be investigated.
We also added the reference 55 in the revised manuscript.
References:
- Crotti, S.; Bottino, N.; Ruggeri, G.M.; Spinelli, E.; Tubiolo, D.; Lissoni, A.; Protti, A.; Gattinoni, L. Spontaneous Breathing during Extracorporeal Membrane Oxygenation in Acute Respiratory Failure. 2017, 126, 678-687.
- Abrams, D.; Schmidt, M.; Pham,T.; Beitler, J.R.; Fan, E.; Goligher, E.C.; McNamee, J.J.; Patroniti, N.; Wilcox, M.E.; Combes, A.; et al. Mechanical Ventilation for Acute Respiratory Distress Syndrome during Extracorporeal Life Support. Research and Practice. Am J Respir Crit Care Med. 2020, 201, 514-525.
- Mauri, T.; Grasselli, G.; Suriano, G.; Eronia, N.; Spadaro, S.; Turrini, C.; Patroniti, N.; Bellani, G.; Pesenti, A. Control of Respiratory Drive and Effort in Extracorporeal Membrane Oxygenation Patients Recovering from Severe Acute Respiratory Distress Syndrome. 2016, 125, 159-67.
We thank the reviewer for valuable comments. Addressing them fully has significantly strengthened the manuscript.
Reviewer 2 Report
Reviewer:
Dear authors,
I read the article entitle: Mechanical Ventilation during Extracorporeal Membrane Oxygenation in Acute Respiratory Distress Syndrome
The review main message is that ECMO can facilitate the recovery in patients ARDS focus on clinical findings, suggestions, protocols, guidelines, and expert opinion.
The review is writing sufficiently, however in my opinion is too optimistic and need to be balance.
Some example: line 20 abstract: …and improve clinical outcome….. this is not completely true.
In COVID-19 ECMO has been improponibile and in some case a disaster.
- Abstract: to be improved.
- Introduction: to be improved and balanced.
- resume all aspect of ventilation setting in a figure or table.
- Discussion: Please discuss also collateral effect of ECMO.
- I never see a VV-ECMO with a FiO2 of 30% and SpO2 sufficiente in acute phase, my be after lung recovery?
- Limitation: Need to state that only hub center have showed reduce mortality.
- Conclusions: please consider to balance your review.
- Best Regards
Author Response
Dear authors,
I read the article entitle: Mechanical Ventilation during Extracorporeal Membrane Oxygenation in Acute Respiratory Distress Syndrome
The review main message is that ECMO can facilitate the recovery in patients ARDS focus on clinical findings, suggestions, protocols, guidelines, and expert opinion.
The review is writing sufficiently, however in my opinion is too optimistic and need to be balance.
Some example: line 20 abstract: …and improve clinical outcome….. this is not completely true.
In COVID-19 ECMO has been improponibile and in some case a disaster.
Response: We thank the reviewer to point out that this article is too optimistic and need to be revised. We agreed with the reviewer that ECMO improve clinical outcomes of ARDS is not completely true. We deleted the statement of “...and improve clinical outcomes”, and added the statement of “However, the collateral effect of ECMO should be considered “in the section of Abstract in the revised manuscript. We also revised the manuscript to be balanced and marked with red words or red text.
Point 1: Abstract: to be improved.
Response 1: We thank the reviewer’s suggestion. Abstract was revised, improved and marked with red text in the revised manuscript.
Point 2: Introduction: to be improved and balanced.
Response 2: We thank the reviewer’s suggestion. The section of introduction was revised, improved and balanced, and marked with red text in the revised manuscript.
Point 3: resume all aspect of ventilation setting in a figure or table.
Response 3: We thank the reviewer’s suggestion to resume all aspects of ventilation settings in a figure or table. We added the Table 3 in the revised manuscript to summarize all aspects of ventilation settings during the initial phase of ECMO for patients with ARDS.
Point 4: Discussion: Please discuss also collateral effect of ECMO.
Response 4: This is an excellent point of view. We appreciated the reviewer’s comment, and agreed that collateral effect of ECMO should be discussed.
Although advances in extracorporeal support techniques, there are many potential collateral effect or complications associated with ECMO and can be devastating. The adverse events included mechanical (i.e., ECMO circuit or device) and medical complications.
The common complications directly related to the ECMO circuit are oxygenator failure, blood clots in oxygenator and other circuit, and other cannula or mechanical related problems (e.g., vessel perforation or dissection, limb ischemia, and incorrect location of cannula).
Bleeding is the most common complication during ECMO because of systemic anticoagulation and thrombocytopenia, and is one of the leading causes of mortality. Other complications not directly related to the ECMO circuit included thromboembolism, hemolysis, disseminated intravascular coagulation, culture-confirmed infection at any site, neurologic injury (e.g., rapid decrease in PaCO2) and renal failure [1, 2].
We added the statement in page 13, the section of “6. Collateral Effect of ECMO” in the revised manuscript as follows (marked with red text):
Although advances in supportive care and innovations in extracorporeal support techniques, there are many potential collateral effect or complications associated with ECMO and can be lethal. Close daily monitoring to minimize the risk of ECMO related complications is necessary, and requires intensive education and training [8]. Therefore, only hub center have showed reduce mortality [68]. The adverse events of ECMO included mechanical (i.e., ECMO circuit or device) and medical complications.
The common complications directly related to the ECMO circuit are oxygenator failure, blood clots in oxygenator and other circuit, and other cannula or mechanical related problems (e.g., vessel perforation or dissection, limb ischemia, and incorrect location of cannula). Bleeding is the most common complication during ECMO because of systemic anticoagulation and thrombocytopenia, and is one of the leading causes of mortality. Other complications not directly related to the ECMO circuit included thromboembolism, hemolysis, disseminated intravascular coagulation, culture-confirmed infection at any site, neurologic injury (e.g., rapid decrease in PaCO2) and renal failure [7, 10].
References:
- Brodie, D.; Bacchetta, M. Extracorporeal membrane oxygenation for ARDS in adults. Engl. J. Med. 2011, 365, 1905–1914.
- Akoumianaki, E.; Jonkman, A.; Sklar, M.C.; Georgopoulos, D.; Brochard, L. A rational approach on the use of extracorporeal membrane oxygenation in severe hypoxemia: advanced technology is not a panacea. Ann Intensive Care. 2021, 11, 107.
Point 5: I never see a VV-ECMO with a FiO2 of 30% and SpO2 sufficiente in acute phase, my be after lung recovery?
Response 5: We thank the reviewer’s comment to point out this issue and we agreed with the reviewer that FiO2 of 30% may not be sufficient in acute and early phases of severe ARDS undergoing ECMO, and is more feasible after lung recovery.
In our experience, the values of FiO2 of mechanical ventilator were around 40–50% during the first 3 days of ECMO.
Although the FiO2 (measured at the ventilator) should be maintained at a low level to reduce the risk of oxygen toxicity and reabsorption atelectasis, higher FiO2 of mechanical ventilator may be necessary if ECMO is inadequate to maintain acceptable levels of oxygenation. The ELSO guideline [1] suggested setting FiO2 at 50% during the first 24 hours, and the values of FiO2 may be decreasing after lung recovery.
We agreed again with the reviewer’s comment that FiO2 of 30% may be feasible after lung recovery.
Reference:
- Extracorporeal Life Support Organization. Available online: https://www.elso.org/Resources/Guidelines.aspx
Point 6: Limitation: Need to state that only hub center have showed reduce mortality.
Response 6: We thank the reviewer’s comment to state that only hub center have showed reduced mortality. We added this statement in page 13, the section of “6. Collateral Effect of ECMO” in the revised manuscript as follows (marked with red text):
Therefore, only hub center have showed reduce mortality [68].
We also added the reference 68.
Reference:
- Barbaro, R.P.; Odetola, F.O.; Kidwell, K.M.; Paden, M.L.; Bartlett, R.H.; Davis, M.M.; Annich, G.M. Association of hospital-level volume of extracorporeal membrane oxygenation cases and mortality. Analysis of the extracorporeal life support organization registry. Am J Respir Crit Care Med. 2015, 191, 894-901.
Point 7: Conclusions: please consider to balance your review.
Response 7: We thank the reviewer’s suggestion. We added the statement of “Collateral Effect of ECMO may be devastating and should be concerned” in the section of Conclusions. We revised the section of conclusions to balance our review and marked with red text in the revised manuscript.
Point 8: Best Regards
Response 8: We thank the reviewer’s valuable comments.
We thank the reviewer for valuable comments. Addressing them fully has significantly strengthened the manuscript.
Round 2
Reviewer 2 Report
Thank you for addressing my comments.
The manuscript now seems to be balance for the reader.
Best Regards